# Submodular Hamming Metrics

**Jennifer Gillenwater**[†]**, Rishabh Iyer**[†]**, Bethany Lusch**[∗]**, Rahul Kidambi**[†]**, Jeff Bilmes**[†]
[†] University of Washington, Dept. of EE, Seattle, U.S.A.
[∗] University of Washington, Dept. of Applied Math, Seattle, U.S.A.
{jengi, rkiyer, herwaldt, rkidambi, bilmes}@uw.edu

## Abstract

We show that there is a largely unexplored class of functions (positive polymatroids) that can define proper discrete metrics over pairs of binary vectors and that are fairly tractable to optimize over. By exploiting submodularity, we are able to give hardness results and approximation algorithms for optimizing over such metrics. Additionally, we demonstrate empirically the effectiveness of these metrics and associated algorithms on both a metric minimization task (a form of clustering) and also a metric maximization task (generating diverse $k$-best lists).

## 1 Introduction

A good distance metric is often the key to an effective machine learning algorithm. For instance, when clustering, the distance metric largely defines which points end up in which clusters. Similarly, in large-margin learning, the distance between different labelings can contribute as much to the definition of the margin as the objective function itself. Likewise, when constructing diverse $k$-best lists, the measure of diversity is key to ensuring meaningful differences between list elements.

We consider distance metrics $d : \{0,1\}^n \times \{0,1\}^n \to \mathbb{R}_+$ over binary vectors, $\boldsymbol{x} \in \{0,1\}^n$. If we define the set $V = \{1, \ldots, n\}$, then each $\boldsymbol{x} = \mathbf{1}_A$ can seen as the characteristic vector of a set $A \subseteq V$, where $\mathbf{1}_A(v) = 1$ if $v \in A$, and $\mathbf{1}_A(v) = 0$ otherwise. For sets $A, B \subseteq V$, with $\triangle$ representing the symmetric difference, $A\triangle B \triangleq (A \setminus B) \cup (B \setminus A)$, the Hamming distance is then: $d_H(A, B) = |A\triangle B| = \sum_{i=1}^n \mathbf{1}_{A\triangle B}(i) = \sum_{i=1}^n \mathbb{1}(\mathbf{1}_A(i) \neq \mathbf{1}_B(i))$. A Hamming distance between two vectors assumes that each entry difference contributes value one. Weighted Hamming distance generalizes this slightly, allowing each entry a unique weight. The Mahalanobis distance further extends this. For many practical applications, however, it is desirable to have entries interact with each other in more complex and higher-order ways than Hamming or Mahalanobis allow. Yet, arbitrary interactions would result in non-metric functions whose optimization would be intractable. In this work, therefore, we consider an alternative class of functions that goes beyond pairwise interactions, yet is computationally feasible, is natural for many applications, and preserves metricity.

Given a set function $f : 2^V \to \mathbb{R}$, we can define a distortion between two binary vectors as follows: $d_f(A, B) = f(A\triangle B)$. By asking $f$ to satisfy certain properties, we will arrive at a class of discrete metrics that is feasible to optimize and preserves metricity. We say that $f$ is *positive* if $f(A) > 0$ whenever $A \neq \emptyset$; $f$ is *normalized* if $f(\emptyset) = 0$; $f$ is *monotone* if $f(A) \leq f(B)$ for all $A \subseteq B \subseteq V$; $f$ is *subadditive* if $f(A) + f(B) \geq f(A \cup B)$ for all $A, B \subseteq V$; $f$ is *modular* if $f(A) + f(B) = f(A \cup B) + f(B \cap A)$ for all $A, B \subseteq V$; and $f$ is *submodular* if $f(A) + f(B) \geq f(A \cup B) + f(B \cap A)$ for all $A, B \subseteq V$. If we assume that $f$ is positive, normalized, monotone, and subadditive then $d_f(A, B)$ is a metric (see Theorem 3.1), but without useful computational properties. If $f$ is positive, normalized, monotone, and modular, then we recover the weighted Hamming distance. In this paper, we assume that $f$ is positive, normalized, monotone, and *sub*modular (and hence also subadditive). These conditions are sufficient to ensure the metricity of $d_f$, but allow for a significant generalization over the weighted Hamming distance. Also, thanks to the properties of submodularity, this class yields efficient optimization algorithms with guarantees

Table 1: Hardness for SH-min and SH-max. UC stands for unconstrained, and Card stands for cardinality-constrained. The entry "open" implies that the problem is potentially poly-time solvable.

| | SH-min | | SH-max | |
|---|---|---|---|---|
| | homogeneous | heterogeneous | homogeneous | heterogeneous |
| UC | Open | $4/3$ | $3/4$ | $3/4$ |
| Card | $\Omega\left(\frac{\sqrt{n}}{1+(\sqrt{n}-1)(1-\kappa_f)}\right)$ | $\Omega\left(\frac{\sqrt{n}}{1+(\sqrt{n}-1)(1-\kappa_f)}\right)$ | $1-1/e$ | $1-1/e$ |

Table 2: Approximation guarantees of algorithms for SH-min and SH-max. '-' implies that no guarantee holds for the corresponding pair. BEST-B only works for the homogeneous case, while all other algorithms work in both cases.

| | UNION-SPLIT | | BEST-B | MAJOR-MIN | RAND-SET |
|---|---|---|---|---|---|
| | UC | Card | UC | Card | UC |
| SH-min | 2 | - | $2-2/m$ | $\frac{n}{1+(n-1)(1-\kappa_f)}$ | - |
| SH-max | $1/4$ | $1/2e$ | - | - | $1/8$ |

for practical machine learning problems. In what follows, we will refer to normalized monotone submodular functions as *polymatroid* functions; all of our results will be concerned with *positive polymatroids*. We note here that despite the restrictions described above, the polymatroid class is in fact quite broad; it contains a number of natural choices of diversity and coverage functions, such as set cover, facility location, saturated coverage, and concave-over-modular functions.

Given a positive polymatroid function $f$, we refer to $d_f(A, B) = f(A \triangle B)$ as a *submodular Hamming (SH) distance*. We study two optimization problems involving these metrics (each $f_i$ is a positive polymatroid, each $B_i \subseteq V$, and $\mathcal{C}$ denotes a combinatorial constraint):

$$\text{SH-min:} \min_{A \in \mathcal{C}} \sum_{i=1}^{m} f_i(A \triangle B_i), \qquad \text{and} \qquad \text{SH-max:} \max_{A \in \mathcal{C}} \sum_{i=1}^{m} f_i(A \triangle B_i). \qquad (1)$$

We will use $\mathcal{F}$ as shorthand for the sequence $(f_1, \ldots, f_m)$, $\mathcal{B}$ for the sequence $(B_1, \ldots, B_m)$, and $F(A)$ for the objective function $\sum_{i=1}^{m} f_i(A \triangle B_i)$. We will also make a distinction between the *homogeneous* case where all $f_i$ are the same function, and the more general *heterogeneous* case where each $f_i$ may be distinct. In terms of constraints, in this paper's theory we consider only the unconstrained ($\mathcal{C} = 2^V$) and the cardinality-constrained (e.g., $|A| \geq k$, $|A| \leq k$) settings. In general though, $\mathcal{C}$ could express more complex concepts such as knapsack constraints, or that solutions must be an independent set of a matroid, or a cut (or spanning tree, path, or matching) in a graph.

Intuitively, the SH-min problem can be thought of as a centroid-finding problem; the minimizing $A$ should be as similar to the $B_i$'s as possible, since a penalty of $f_i(A \triangle B_i)$ is paid for each difference. Analogously, the SH-max problem can be thought of as a diversification problem; the maximizing $A$ should be as distinct from all $B_i$'s as possible, as $f_i(A \triangle B)$ is awarded for each difference. Given modular $f_i$ (the weighted Hamming distance case), these optimization problems can be solved exactly and efficiently for many constraint types. For the more general case of submodular $f_i$, we establish several hardness results and offer new approximation algorithms, as summarized in Tables 1 and 2. Our main contribution is to provide (to our knowledge), the first systematic study of the properties of submodular Hamming (SH) metrics, by showing metricity, describing potential machine learning applications, and providing optimization algorithms for SH-min and SH-max.

The outline of this paper is as follows. In Section 2, we offer further motivation by describing several applications of SH-min and SH-max to machine learning. In Section 3, we prove that for a positive polymatroid function $f$, the distance $d_f(A, B) = f(A \triangle B)$ is a metric. Then, in Sections 4 and 5 we give hardness results and approximation algorithms, and in Section 6 we demonstrate the practical advantage that submodular metrics have over modular metrics for several real-world applications.

## 2 Applications

We motivate SH-min and SH-max by showing how they occur naturally in several applications.

**Clustering**: Many clustering algorithms, including for example $k$-means [1], use distance functions in their optimization. If each item $i$ to be clustered is represented by a binary feature vector $\mathbf{b}_i \in \{0,1\}^n$, then counting the disagreements between $\mathbf{b}_i$ and $\mathbf{b}_j$ is one natural distance function. Defining sets $B_i = \{v : \mathbf{b}_i(v) = 1\}$, this count is equivalent to the Hamming distance $|B_i \triangle B_j|$. Consider a document clustering application where $V$ is the set of all features (e.g., $n$-grams) and $B_i$ is the set of features for document $i$. Hamming distance has value 2 both when $B_i \triangle B_j = \{\text{"submodular", "synapse"}\}$ and when $B_i \triangle B_j = \{\text{"submodular", "modular"}\}$. Intuitively, however, a smaller distance seems warranted in the latter case since the difference is only in one rather than two distinct concepts. The submodular Hamming distances we propose in this work can easily capture this type of behavior. Given feature clusters $\mathcal{W}$, one can define a submodular function as: $f(Y) = \sum_{W \in \mathcal{W}} \sqrt{|Y \cap W|}$. Applying this with $Y = B_i \triangle B_j$, if the documents' differences are confined to one cluster, the distance is smaller than if the differences occur across several word clusters. In the case discussed above, the distances are 2 and $\sqrt{2}$. If this submodular Hamming distance is used for $k$-means clustering, then the mean-finding step becomes an instance of the SH-min problem. That is, if cluster $j$ contains documents $C_j$, then its mean takes exactly the following SH-min form: $\mu_j \in \operatorname{argmin}_{A \subseteq V} \sum_{i \in C_j} f(A \triangle B_i)$.

**Structured prediction**: Structured support vector machines (SVMs) typically rely on Hamming distance to compare candidate structures to the true one. The margin required between the correct structure score and a candidate score is then proportional to their Hamming distance. Consider the problem of segmenting an image into foreground and background. Let $B_i$ be image $i$'s true set of foreground pixels. Then Hamming distance between $B_i$ and a candidate segmentation with foreground pixels $A$ counts the number of mis-labeled pixels. However, both [2] and [3] observe poor performance with Hamming distance and recent work by [4] shows improved performance with richer distances that are supermodular functions of $A$. One potential direction for further enriching image segmentation distance functions is thus to consider non-modular functions from within our submodular Hamming metrics class. These functions have the ability to correct for the over-penalization that the current distance functions may suffer from when the same kind of difference happens repeatedly. For instance, if $B_i$ differs from $A$ only in the pixels local to a particular block of the image, then current distance functions could be seen as over-estimating the difference. Using a submodular Hamming function, the "loss-augmented inference" step in SVM optimization becomes an SH-max problem. More concretely, if the segmentation model is defined by a submodular graph cut $g(A)$, then we have: $\max_{A \subseteq V} g(A) + f(A \triangle B_i)$. (Note that $g(A) = g(A \triangle \emptyset)$.) In fact, [5] observes superior results with this type of loss-augmented inference using a special case of a submodular Hamming metric for the task of multi-label image classification.

**Diverse $k$-best**: For some machine learning tasks, rather than finding a model's single highest-scoring prediction, it is helpful to find a diverse set of high-quality predictions. For instance, [6] showed that for image segmentation and pose tracking a diverse set of $k$ solutions tended to contain a better predictor than the top $k$ highest-scoring solutions. Additionally, finding diverse solutions can be beneficial for accommodating user interaction. For example, consider the task of selecting 10 photos to summarize the 100 photos that a person took while on vacation. If the model's best prediction (a set of 10 images) is rejected by the user, then the system should probably present a substantially different prediction on its second try. Submodular functions are a natural model for several summarization problems [7, 8]. Thus, given a submodular summarization model $g$, and a set of existing diverse summaries $A_1, A_2, \ldots, A_{k-1}$, one could find a $k$th summary to present to the user by solving: $A_k = \operatorname{argmax}_{A \subseteq V, |A| = \ell} g(A) + \sum_{i=1}^{k-1} f(A \triangle A_i)$. If $f$ and $g$ are both positive polymatroids, then this constitutes an instance of the SH-max problem.

## 3  Properties of the submodular Hamming metric

We next show several interesting properties of the submodular Hamming distance. Proofs for all theorems and lemmas can be found in the supplementary material. We begin by showing that any positive polymatroid function of $A \triangle B$ is a metric. In fact, we show the more general result that any positive normalized monotone subadditive function of $A \triangle B$ is a metric. This result is known (see for instance Chapter 8 of [9]), but we provide a proof (in the supplementary material) for completeness.

**Theorem 3.1.** *Let $f : 2^V \to \mathbb{R}$ be a positive normalized monotone subadditive function. Then $d_f(A, B) = f(A \triangle B)$ is a metric on $A, B \subseteq V$.*

While these subadditive functions are metrics, their optimization is known to be very difficult. The simple subadditive function example in the introduction of [10] shows that subadditive minimization is inapproximable, and Theorem 17 of [11] states that no algorithm exists for subadditive maximization that has an approximation factor better than $\tilde{O}(\sqrt{n})$. By contrast, submodular minimization is poly-time in the unconstrained setting [12], and a simple greedy algorithm from [13] gives a $1 - 1/e$-approximation for maximization of positive polymatroids subject to a cardinality constraint. Many other approximation results are also known for submodular function optimization subject to various other types of constraints. Thus, in this work we restrict ourselves to positive polymatroids.

**Corollary 3.1.1.** *Let $f : 2^V \to \mathbb{R}_+$ be a positive polymatroid function. Then $d_f(A, B) = f(A\triangle B)$ is a metric on $A, B \subseteq V$.*

This restriction does not entirely resolve the question of optimization hardness though. Recall that the optimization in SH-min and SH-max is with respect to $A$, but that the $f_i$ are applied to the sets $A\triangle B_i$. Unfortunately, the function $g_B(A) = f(A\triangle B)$, for a fixed set $B$, is neither necessarily submodular nor supermodular in $A$. The next example demonstrates this violation of submodularity.

**Example 3.1.1.** *To be submodular, the function $g_B(A) = f(A\triangle B)$ must satisfy the following condition for all sets $A_1, A_2 \subseteq V$: $g_B(A_1) + g_B(A_2) \geq g_B(A_1 \cup A_2) + g_B(A_1 \cap A_2)$. Consider the positive polymatroid function $f(Y) = \sqrt{|Y|}$ and let $B$ consist of two elements: $B = \{b_1, b_2\}$. Then for $A_1 = \{b_1\}$ and $A_2 = \{c\}$ (with $c \notin B$): $g_B(A_1) + g_B(A_2) = \sqrt{1} + \sqrt{3} < 2\sqrt{2} = g_B(A_1 \cup A_2) + g_B(A_1 \cap A_2)$.*

Although $g_B(A) = f(A\triangle B)$ can be non-submodular, we are interestingly still able to make use of the fact that $f$ is submodular in $A\triangle B$ to develop approximation algorithms for SH-min and SH-max.

# 4 Minimization of the submodular Hamming metric

In this section, we focus on SH-min (the centroid-finding problem). We consider the four cases from Table 1: the constrained ($A \in \mathcal{C} \subset 2^V$) and unconstrained ($A \in \mathcal{C} = 2^V$) settings, as well as the homogeneous case (where all $f_i$ are the same function) and the heterogeneous case. Before diving in, we note that in all cases we assume not only the natural oracle access to the objective function $F(A) = \sum_{i=1}^{m} f_i(A\triangle B_i)$ (i.e., the ability to evaluate $F(A)$ for any $A \subseteq V$), but also knowledge of the $B_i$ (the $\mathcal{B}$ sequence). Theorem 4.1 shows that without knowledge of $\mathcal{B}$, SH-min is inapproximable. In practice, requiring knowledge of $\mathcal{B}$ is not a significant limitation; for all of the applications described in Section 2, $\mathcal{B}$ is naturally known.

**Theorem 4.1.** *Let $f$ be a positive polymatroid function. Suppose that the subset $B \subseteq V$ is fixed but unknown and $g_B(A) = f(A\triangle B)$. If we only have an oracle for $g_B$, then there is no poly-time approximation algorithm for minimizing $g_B$, up to any polynomial approximation factor.*

## 4.1 Unconstrained setting

Submodular minimization is poly-time in the unconstrained setting [12]. Since a sum of submodular functions is itself submodular, at first glance it might then seem that the sum of $f_i$ in SH-min can be minimized in poly-time. However, recall from Example 3.1.1 that the $f_i$'s are not necessarily submodular in the optimization variable, $A$. This means that the question of SH-min's hardness, even in the unconstrained setting, is an open question. Theorem 4.2 resolves this question for the heterogeneous case, showing that it is NP-hard and that no algorithm can do better than a $4/3$-approximation guarantee. The question of hardness in the homogeneous case remains open.

**Theorem 4.2.** *The unconstrained and heterogeneous version of SH-min is NP-hard. Moreover, no poly-time algorithm can achieve an approximation factor better than $4/3$.*

Since unconstrained SH-min is NP-hard, it makes sense to consider approximation algorithms for this problem. We first provide a simple 2-approximation, UNION-SPLIT (see Algorithm 1). This algorithm splits $f(A\triangle B) = f((A \setminus B) \cup (B \setminus A))$ into $f(A \setminus B) + f(B \setminus A)$, then applies standard submodular minimization (see e.g. [14]) to the split function. Theorem 4.3 shows that this algorithm is a 2-approximation for SH-min. It relies on Lemma 4.2.1, which we state first.

**Lemma 4.2.1.** *Let $f$ be a positive monotone subadditive function. Then, for any $A, B \subseteq V$:*

$$f(A\triangle B) \leq f(A \setminus B) + f(B \setminus A) \leq 2f(A\triangle B). \tag{2}$$

**Algorithm 1** UNION-SPLIT

> **Input**: $\mathcal{F}, \mathcal{B}$
> Define $f_i'(Y) = f_i(Y \setminus B_i) + f_i(B_i \setminus Y)$
> Define $F'(Y) = \sum_{i=1}^{m} f_i'(Y)$
> **Output**: SUBMODULAR-OPT $(F')$

**Algorithm 2** BEST-B

> **Input**: $F, \mathcal{B}$
> $A \leftarrow B_1$
> **for** $i = 2, \ldots, m$ **do**
>     **if** $F(B_i) < F(A)$: $A \leftarrow B_i$
> **Output**: $A$

**Algorithm 3** MAJOR-MIN

> **Input**: $\mathcal{F}, \mathcal{B}, \mathcal{C}$
> $A \leftarrow \emptyset$
> **repeat**
>     $c \leftarrow F(A)$
>     Set $\boldsymbol{w}_{\hat{F}}$ as in Equation 3
>     $A \leftarrow$ MODULAR-MIN $(\boldsymbol{w}_{\hat{F}}, \mathcal{C})$
> **until** $F(A) = c$
> **Output**: $A$

**Theorem 4.3.** UNION-SPLIT *is a* 2-*approximation for unconstrained SH-min.*

Restricting to the homogeneous setting, we can provide a different algorithm that has a better approximation guarantee than UNION-SPLIT. This algorithm simply checks the value of $F(A) = \sum_{i=1}^{m} f(A \triangle B_i)$ for each $B_i$ and returns the minimizing $B_i$. We call this algorithm BEST-B (Algorithm 2). Theorem 4.4 gives the approximation guarantee for BEST-B. This result is known [15], as the proof of the guarantee only makes use of metricity and homogeneity (not submodularity), and these properties are common to much other work. We provide the proof in our notation for completeness though.

**Theorem 4.4.** *For* $m = 1$, BEST-B *exactly solves unconstrained SH-min. For* $m > 1$, BEST-B *is a* $\left(2 - \frac{2}{m}\right)$-*approximation for unconstrained homogeneous SH-min.*

## 4.2 Constrained setting

In the constrained setting, the SH-min problem becomes more difficult. Essentially, all of the hardness results established in existing work on constrained submodular minimization applies to the constrained SH-min problem as well. Theorem 4.5 shows that, even for a simple cardinality constraint and identical $f_i$ (homogeneous setting), not only is SH-min NP-hard, but also it is hard to approximate with a factor better than $\Omega(\sqrt{n})$.

**Theorem 4.5.** *Homogeneous SH-min is NP-hard under cardinality constraints. Moreover, no algorithm can achieve an approximation factor better than* $\Omega\left(\frac{\sqrt{n}}{1+(\sqrt{n}-1)(1-\kappa_f)}\right)$, *where* $\kappa_f = 1 - \min_{j \in V} \frac{f(j|V \setminus j)}{f(j)}$ *denotes the curvature of* $f$. *This holds even when* $m = 1$.

We can also show similar hardness results for several other combinatorial constraints including matroid constraints, shortest paths, spanning trees, cuts, etc. [16, 17]. Note that the hardness established in Theorem 4.5 depends on a quantity $\kappa_f$, which is also called the *curvature* of a submodular function [18, 16]. Intuitively, this factor measures how close a submodular function is to a modular function. The result suggests that the closer the function is being modular, the easier it is to optimize. This makes sense, since with a modular function, SH-min can be exactly minimized under several combinatorial constraints. To see this for the cardinality-constrained case, first note that for modular $f_i$, the corresponding $F$-function is also modular. Lemma 4.5.1 formalizes this.

**Lemma 4.5.1.** *If the* $f_i$ *in SH-min are modular, then* $F(A) = \sum_{i=1}^{m} f_i(A \triangle B_i)$ *is also modular.*

Given Lemma 4.5.1, from the definition of modularity we know that there exists some constant $C$ and vector $\boldsymbol{w}_F \in \mathbb{R}^n$, such that $F(A) = C + \sum_{j \in A} w_F(j)$. From this representation it is clear that $F$ can be minimized subject to the constraint $|A| \geq k$ by choosing as the set $A$ the items corresponding to the $k$ smallest entries in $w_F$. Thus, for modular $f_i$, or $f_i$ with small curvature $\kappa_{f_i}$, such constrained minimization is relatively easy.

Having established the hardness of constrained SH-min, we now turn to considering approximation algorithms for this problem. Unfortunately, the UNION-SPLIT algorithm from the previous section

requires an efficient algorithm for submodular function minimization, and no such algorithm exists in the constrained setting; submodular minimization is NP-hard even under simple cardinality constraints [19]. Similarly, the BEST-B algorithm breaks down in the constrained setting; its guarantees carry over only if all the $B_i$ are within the constraint set $\mathcal{C}$. Thus, for the constrained SH-min problem we instead propose a majorization-minimization algorithm. Theorem 4.6 shows that this algorithm has an $O(n)$ approximation guarantee, and Algorithm 3 formally defines the algorithm.

Essentially, MAJOR-MIN proceeds by iterating the following two steps: constructing $\hat{F}$, a modular upper bound for $F$ at the current solution $A$, then minimizing $\hat{F}$ to get a new $A$. $\hat{F}$ consists of superdifferentials [20, 21] of $F$'s component submodular functions. We use the superdifferentials defined as "grow" and "shrink" in [22]. Defining sets $S, T$ as $S = V \setminus j, T = A \triangle B_i$ for "grow", and $S = (A \triangle B_i) \setminus j, T = \emptyset$ for "shrink", the $\boldsymbol{w}_{\hat{F}}$ vector that represents the modular $\hat{F}$ can be written:

$$w_{\hat{F}}(j) = \sum_{i=1}^{m} \begin{cases} f_i(j \mid S) \text{ if } j \in A \triangle B_i \\ f_i(j \mid T) \text{ otherwise,} \end{cases} \tag{3}$$

where $f(Y \mid X) = f(Y \cup X) - f(X)$ is the gain in $f$-value when adding $Y$ to $X$. We now state the main theorem characterizing algorithm MAJOR-MIN's performance on SH-min.

**Theorem 4.6.** MAJOR-MIN *is guaranteed to improve the objective value,* $F(A) = \sum_{i=1}^{m} f_i(A \triangle B_i)$, *at every iteration. Moreover, for any constraint over which a modular function can be exactly optimized, it has a* $\left( \max_i \frac{|A^* \triangle B_i|}{1 + (|A^* \triangle B_i| - 1)(1 - \kappa_{f_i}(A^* \triangle B_i))} \right)$ *approximation guarantee, where* $A^*$ *is the optimal solution of SH-min.*

While MAJOR-MIN does not have a constant-factor guarantee (which is possible only in the unconstrained setting), the bounds are not too far from the hardness of the constrained setting. For example, in the cardinality case, the guarantee of MAJOR-MIN is $\frac{n}{1+(n-1)(1-\kappa_f)}$, while the hardness shown in Theorem 4.5 is $\Omega\left(\frac{\sqrt{n}}{1+(n-1)(1-\kappa_f)}\right)$.

# 5 Maximization of the submodular Hamming metric

We next characterize the hardness of SH-max (the diversification problem) and describe approximation algorithms for it. We first show that all versions of SH-max, even the unconstrained homogeneous one, are NP-hard. Note that this is a non-trivial result. Maximization of a monotone function such as a polymatroid is not NP-hard; the maximizer is always the full set $V$. But, for SH-max, despite the fact that the $f_i$ are monotone with respect to their argument $A \triangle B_i$, they are not monotone with respect to $A$ itself. This makes SH-max significantly harder. After establishing that SH-max is NP-hard, we show that no poly-time algorithm can obtain an approximation factor better $3/4$ in the unconstrained setting, and a factor of $(1 - 1/e)$ in the constrained setting. Finally, we provide a simple approximation algorithm which achieves a factor of $1/4$ for all settings.

**Theorem 5.1.** *All versions of SH-max (constrained or unconstrained, heterogeneous or homogeneous) are NP-hard. Moreover, no poly-time algorithm can obtain a factor better than* $3/4$ *for the unconstrained versions, or better than* $1 - 1/e$ *for the cardinality-constrained versions.*

We turn now to approximation algorithms. For the unconstrained setting, Lemma 5.1.1 shows that simply choosing a random subset, $A \subseteq V$ provides a $1/8$-approximation in expectation.

**Lemma 5.1.1.** *A random subset is a* $1/8$-*approximation for SH-max in the unconstrained (homogeneous or heterogeneous) setting.*

An improved approximation guarantee of $1/4$ can be shown for a variant of UNION-SPLIT (Algorithm 1), if the call to SUBMODULAR-OPT is a call to a SUBMODULAR-MAX algorithm. Theorem 5.2 makes this precise for both the unconstrained case and a cardinality-constrained case. It might also be of interest to consider more complex constraints, such as matroid independence and base constraints, but we leave the investigation of such settings to future work.

**Theorem 5.2.** *Maximizing* $\bar{F}(A) = \sum_{i=1}^{m} \left( f_i(A \setminus B_i) + f_i(B_i \setminus A) \right)$ *with a bi-directional greedy algorithm [23, Algorithm 2] is a linear-time* $1/4$-*approximation for maximizing* $F(A) = \sum_{i=1}^{m} f_i(A \triangle B_i)$, *in the unconstrained setting. Under the cardinality constraint* $|A| \leq k$, *using the randomized greedy algorithm [24, Algorithm 1] provides a* $\frac{1}{2e}$-*approximation.*

Table 3: mV-ROUGE averaged over the 14 datasets ($\pm$ standard deviation).

| | HM | SP | TP |
|---|---|---|---|
| | $0.38 \pm 0.14$ | $0.43 \pm 0.20$ | $\mathbf{0.50 \pm 0.26}$ |

Table 4: # of wins (out of 14 datasets).

| HM | SP | TP |
|---|---|---|
| 3 | 1 | **10** |

## 6 Experiments

To demonstrate the effectiveness of the submodular Hamming metrics proposed here, we apply them to a metric minimization task (clustering) and a metric maximization task (diverse $k$-best).

### 6.1 SH-min application: clustering

We explore the document clustering problem described in Section 2, where the groundset $V$ is all unigram features and $B_i$ contains the unigrams of document $i$. We run $k$-means clustering and each iteration find the mean for cluster $C_j$ by solving: $\mu_j \in \operatorname{argmin}_{A:|A|\geq\ell} \sum_{i\in C_j} f(A \triangle B_i)$. The constraint $|A| \geq \ell$ requires the mean to contain at least $\ell$ unigrams, which helps $k$-means to create richer and more meaningful cluster centers. We compare using the submodular function $f(Y) = \sum_{W \in \mathcal{W}} \sqrt{|Y \cap W|}$ (SM), to using Hamming distance (HM). The problem of finding $\mu_j$ above can be solved exactly for HM, since it is a modular function. In the SM case, we apply MAJOR-MIN (Algorithm 3). As an initial test, we generate synthetic data consisting of 100 "documents" assigned to 10 "true" clusters. We set the number of "word" features to $n = 1000$, and partition the features into 100 word classes (the $\mathcal{W}$ in the submodular function). Ten word classes are associated with each true document cluster, and each document contains one word from each of these word classes. That is, each word is contained in only one document, but documents in the same true cluster have words from the same word classes. We set the minimum cluster center size to $\ell = 100$. We use $k$-means++ initialization [25] and average over 10 trials. Within the $k$-means optimization, we enforce that all clusters are of equal size by assigning a document to the closest center whose current size is $< 10$. With this setup, the average accuracy of HM is 28.4% ($\pm 2.4$), while SM is 69.4% ($\pm 10.5$). The HM accuracy is essentially the accuracy of a random assignment of documents to clusters; this makes sense, as no documents share words, rendering the Hamming distance useless. In real-world data there would likely be some word overlap though; to better model this, we let each document contain a random sampling of 10 words from the word clusters associated with its document cluster. In this case, the average accuracy of HM is 57.0% ($\pm 6.8$), while SM is 88.5% ($\pm 8.4$). The results for SM are even better if randomization is removed from the initialization (we simply choose the next center to be one with greatest distance from the current centers). In this case, the average accuracy of HM is 56.7% ($\pm 7.1$), while SM is 100% ($\pm 0.0$). This indicates that as long as the starting point for SM contains one document from each cluster, the SM optimization will recover the true clusters.

Moving beyond synthetic data, we applied the same method to the problem of clustering NIPS papers. The initial set of documents that we consider consists of all NIPS papers[1] from 1987 to 2014. We filter the words of a given paper by first removing stopwords and any words that don't appear at least 3 times in the paper. We further filter by removing words that have small tf-idf value ($< 0.001$) and words that occur in only one paper or in more than 10% of papers. We then filter the papers themselves, discarding any that have fewer than 25 remaining words and for each other paper retaining only its top (by tf-idf score) 25 words. Each of the 5,522 remaining papers defines a $B_i$ set. Among the $B_i$ there are 12,262 unique words. To get the word clusters $\mathcal{W}$, we first run the WORD2VEC code of [26], which generates a 100-dimensional real-valued vector of features for each word, and then run $k$-means clustering with Euclidean distance on these vectors to define 100 word clusters. We set the center size cardinality constraint to $\ell = 100$ and set the number of document clusters to $k = 10$. To initialize, we again use $k$-means++ [25], with $k = 10$. Results are averaged over 10 trials. While we do not have groundtruth labels for NIPS paper clusters, we can use within-cluster distances as a proxy for cluster goodness (lower values, indicating tighter clusters, are better). Specifically, we compute: $k$-means-score $= \sum_{j=1}^{k} \sum_{i \in C_j} g(\mu_j \triangle B_i)$. With Hamming for $g$, the average ratio of HM's $k$-means-score to SM's is $0.916 \pm 0.003$. This indicates that, as expected, HM does a better job of optimizing the Hamming loss. However, with the submodular function for $g$, the average ratio of HM's $k$-means-score to SM's is $1.635 \pm 0.038$. Thus, SM does a significantly better job optimizing the submodular loss.

## 6.2 SH-max application: diverse $k$-best

In this section, we explore a diverse $k$-best image collection summarization problem, as described in Section 2. For this problem, our goal is to obtain $k$ summaries, each of size $l$, by selecting from a set consisting of $n \gg l$ images. The idea is that either: (a) the user could choose from among these $k$ summaries the one that they find most appealing, or (b) a (more computationally expensive) model could be applied to re-rank these $k$ summaries and choose the best. As is described in Section 2, we obtain the $k$th summary $A_k$, given the first $k-1$ summaries $A_{1:k-1}$ via: $A_k = \operatorname{argmax}_{A \subseteq V, |A|=\ell} g(A) + \sum_{i=1}^{k-1} f(A \triangle A_i)$.

For $g$ we use the facility location function: $g(A) = \sum_{i \in V} \max_{j \in A} S_{ij}$, where $S_{ij}$ is a similarity score for images $i$ and $j$. We compute $S_{ij}$ by taking the dot product of the $i$th and $j$th feature vectors, which are the same as those used by [8]. For $f$ we compare two different functions: (1) $f(A \triangle A_i) = |A \triangle A_i|$, the Hamming distance (HM), and (2) $f(A \triangle A_i) = g(A \triangle A_i)$, the submodular facility location distance (SM). For HM we optimize via the standard greedy algorithm [13]; since the facility location function $g$ is monotone submodular, this implies an approximation guarantee of $(1 - 1/e)$. For SM, we experiment with two algorithms: (1) standard greedy [13], and (2) UNION-SPLIT (Algorithm 1) with standard greedy as the SUBMODULAR-OPT function. We will refer to these two cases as "single part" (SP)

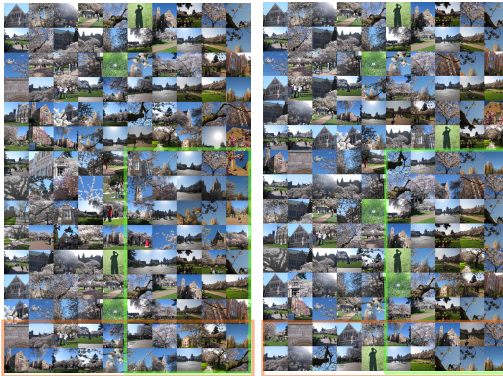

Figure 1: An example photo montage (zoom in to see detail) showing 15 summaries of size 10 (one per row) from the HM approach (left) and the TP approach (right), for image collection #6.

and "two part" (TP). Note that neither of these optimization techniques has a formal approximation guarantee, though the latter would if instead of standard greedy we used the bi-directional greedy algorithm of [23]. We opt to use standard greedy though, as it typically performs much better in practice. We employ the image summarization dataset from [8], which consists of 14 image collections, each of which contains $n = 100$ images. For each image collection, we seek $k = 15$ summaries of size $\ell = 10$. For evaluation, we employ the V-ROUGE score developed by [8]; the mean V-ROUGE (mV-ROUGE) of the $k$ summaries provides a quantitative measure of their goodness. V-ROUGE scores are normalized such that a score of 0 corresponds to randomly generated summaries, while a score of 1 is on par with human-generated summaries.

Table 3 shows that SP and TP outperform HM in terms of mean mV-ROUGE, providing support for the idea of using submodular Hamming distances in place of (modular) Hamming for diverse $k$-best applications. TP also outperforms SP, suggesting that the objective-splitting used in UNION-SPLIT is of practical significance. Table 4 provides additional evidence of TP's superiority, indicating that for 10 out of the 14 image collections, TP has the best mV-ROUGE score of the three approaches.

Figure 1 provides some qualitative evidence of TP's goodness. Notice that the images in the green rectangle tend to be more redundant with images from the previous summaries in the HM case than in the TP case; the HM solution contains many images with a "sky" theme, while TP contains more images with other themes. This shows that the HM solution lacks diversity across summaries. The quality of the individual summaries also tends to become poorer for the later HM sets; considering the images in the red rectangles overlaid on the montage, the HM sets contain many images of tree branches here. By contrast, the TP summary quality remains good even for the last few summaries.

## 7 Conclusion

In this work we defined a new class of distance functions: submodular Hamming metrics. We established hardness results for the associated SH-min and SH-max problems, and provided approximation algorithms. Further, we demonstrated the practicality of these metrics for several applications. There remain several open theoretical questions (e.g., the tightness of the hardness results and the NP-hardness of SH-min), as well as many opportunities for applying submodular Hamming metrics to other machine learning problems (e.g., the prediction application from Section 2).

## Footnotes

[1]Papers were downloaded from http://papers.nips.cc/.

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
