[Supplementary Material]

# Submodular Hamming Metrics

**Jennifer Gillenwater**[†]**, Rishabh Iyer**[†]**, Bethany Lusch**[∗]**, Rahul Kidambi**[†]**, Jeff Bilmes**[†]
[†] University of Washington, Dept. of EE, Seattle, U.S.A.
[∗] University of Washington, Dept. of Applied Math, Seattle, U.S.A.
{jengi, rkiyer, herwaldt, rkidambi, bilmes}@uw.edu

## Abstract

We show that there is a largely unexplored class of functions (positive polymatroids) that can define proper discrete metrics over pairs of binary vectors and that are fairly tractable to optimize over. By exploiting submodularity, we are able to give hardness results and approximation algorithms for optimizing over such metrics. Additionally, we demonstrate empirically the effectiveness of these metrics and associated algorithms on both a metric minimization task (a form of clustering) and also a metric maximization task (generating diverse $k$-best lists).

## 1 Introduction

A good distance metric is often the key to an effective machine learning algorithm. For instance, when clustering, the distance metric largely defines which points end up in which clusters. Similarly, in large-margin learning, the distance between different labelings can contribute as much to the definition of the margin as the objective function itself. Likewise, when constructing diverse $k$-best lists, the measure of diversity is key to ensuring meaningful differences between list elements.

We consider distance metrics $d : \{0,1\}^n \times \{0,1\}^n \to \mathbb{R}_+$ over binary vectors, $\boldsymbol{x} \in \{0,1\}^n$. If we define the set $V = \{1, \ldots, n\}$, then each $\boldsymbol{x} = \mathbf{1}_A$ can seen as the characteristic vector of a set $A \subseteq V$, where $\mathbf{1}_A(v) = 1$ if $v \in A$, and $\mathbf{1}_A(v) = 0$ otherwise. For sets $A, B \subseteq V$, with $\triangle$ representing the symmetric difference, $A \triangle B \triangleq (A \setminus B) \cup (B \setminus A)$, the Hamming distance is then:

$$d_H(A, B) = |A \triangle B| = \sum_{i=1}^{n} \mathbf{1}_{A \triangle B}(i) = \sum_{i=1}^{n} \mathbb{1}(\mathbf{1}_A(i) \neq \mathbf{1}_B(i)). \tag{1}$$

A Hamming distance between two vectors assumes that each entry difference contributes value one. Weighted Hamming distance generalizes this slightly, allowing each entry a unique weight. Mahalanobis distance generalizes further, allowing weighted pairwise interactions of the following form:

$$d_M(A, B) = \mathbf{1}_{A \triangle B}^{\top} S \mathbf{1}_{A \triangle B} = \sum_{i=1}^{n} \sum_{j=1}^{n} S_{ij} \mathbf{1}_{A \triangle B}(i) \mathbf{1}_{A \triangle B}(j). \tag{2}$$

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

*Proof.* Let $A, B, C \subseteq V$ be arbitrary. We check each of the four properties of metrics:

1. Proof of non-negativity: $d(A, B) = f(A \triangle B) \geq 0$ because $f$ is normalized and positive.

2. Proof of identity of indiscernibles: $d(A, B) = 0 \Leftrightarrow f(A \triangle B) = 0 \Leftrightarrow A \triangle B = \emptyset \Leftrightarrow A = B$. The third implication follows because of normalization and positivity of $f$, and the fourth follows from the definition of $\triangle$.

3. Proof of symmetry: $d(A, B) = f(A \triangle B) = f(B \triangle A) = d(B, A)$, by definition of $\triangle$.

4. Proof of the triangle inequality: First, note that $A \triangle B \subseteq (A \triangle C) \cup (C \triangle B)$. This follows because each element $v \in A \setminus B$ is either in $C \setminus B$ (true if $v \in C$) or in $A \setminus C$ (true if $v \notin C$). Similarly, each element $v \in B \setminus A$ is either in $C \setminus A$ (true if $v \in C$) or in $B \setminus C$ (true if $v \notin C$). Then, because $f$ is monotone and subadditive, we have:
$$f(A \triangle B) \leq f((A \triangle C) \cup (C \triangle B)) \leq f(A \triangle C) + f(C \triangle B). \qquad (7)$$

$\square$

While these subadditive functions are metrics, their optimization is known to be very difficult. The simple subadditive function example in the introduction of [10] shows that subadditive minimization is inapproximable, and Theorem 17 of [11] states that no algorithm exists for subadditive maximization that has an approximation factor better than $\tilde{O}(\sqrt{n})$. By contrast, submodular minimization is poly-time in the unconstrained setting [12], and a simple greedy algorithm from [13] gives a $1 - 1/e$-approximation for maximization of positive polymatroids subject to a cardinality constraint. Many other approximation results are also known for submodular function optimization subject to various other types of constraints. Thus, in this work we restrict ourselves to positive polymatroids.

**Corollary 3.1.1.** *Let $f : 2^V \to \mathbb{R}_+$ be a positive polymatroid function. Then $d_f(A, B) = f(A \triangle B)$ is a metric on $A, B \subseteq V$.*

This restriction does not entirely resolve the question of optimization hardness though. Recall that the optimization in SH-min and SH-max is with respect to $A$, but that the $f_i$ are applied to the sets $A \triangle B_i$. Unfortunately, the function $g_B(A) = f(A \triangle B)$, for a fixed set $B$, is neither necessarily submodular nor supermodular in $A$. The next example demonstrates this violation of submodularity.

**Example 3.1.1.** *To be submodular, the function $g_B(A) = f(A \triangle B)$ must satisfy the following condition for all sets $A_1, A_2 \subseteq V$: $g_B(A_1) + g_B(A_2) \geq g_B(A_1 \cup A_2) + g_B(A_1 \cap A_2)$. Consider the positive polymatroid function $f(Y) = \sqrt{|Y|}$ and let $B$ consist of two elements: $B = \{b_1, b_2\}$. Then for $A_1 = \{b_1\}$ and $A_2 = \{c\}$ (with $c \notin B$):*
$$g_B(A_1) + g_B(A_2) = \sqrt{1} + \sqrt{3} < 2\sqrt{2} = g_B(A_1 \cup A_2) + g_B(A_1 \cap A_2). \qquad (8)$$
*This violates the definition of submodularity, implying that $g_B(A)$ is not submodular.*

Although $g_B(A) = f(A \triangle B)$ can be non-submodular, we are interestingly still able to make use of the fact that $f$ is submodular in $A \triangle B$ to develop approximation algorithms for SH-min and SH-max.

# 4 Minimization of the submodular Hamming metric

In this section, we focus on SH-min (the centroid-finding problem). We consider the four cases from Table 1: the constrained ($A \in \mathcal{C} \subset 2^V$) and unconstrained ($A \in \mathcal{C} = 2^V$) settings, as well as the homogeneous case (where all $f_i$ are the same function) and the heterogeneous case. Before diving in, we note that in all cases we assume not only the natural oracle access to the objective function $F(A) = \sum_{i=1}^{m} f_i(A \triangle B_i)$ (i.e., the ability to evaluate $F(A)$ for any $A \subseteq V$), but also knowledge of the $B_i$ (the $\mathcal{B}$ sequence). Theorem 4.1 shows that without knowledge of $\mathcal{B}$, SH-min is inapproximable. In practice, requiring knowledge of $\mathcal{B}$ is not a significant limitation; for all of the applications described in Section 2, $\mathcal{B}$ is naturally known.

**Theorem 4.1.** *Let $f$ be a positive polymatroid function. Suppose that the subset $B \subseteq V$ is fixed but unknown and $g_B(A) = f(A \triangle B)$. If we only have an oracle for $g_B$, then there is no poly-time approximation algorithm for minimizing $g_B$, up to any polynomial approximation factor.*

*Proof.* Define $f(Y)$ as follows:

$$f(Y) = \begin{cases} 0 & \text{if } Y = \emptyset \\ 1 & \text{otherwise.} \end{cases} \tag{9}$$

Then $g_B(A) = 1$ unless $A = B$. Thus, it would take any algorithm an exponential number of queries on $g_B$ to find $B$. $\square$

## 4.1 Unconstrained setting

Submodular minimization is poly-time in the unconstrained setting [12]. Since a sum of submodular functions is itself submodular, at first glance it might then seem that the sum of $f_i$ in SH-min can be minimized in poly-time. However, recall from Example 3.1.1 that the $f_i$'s are not necessarily submodular in the optimization variable, $A$. This means that the question of SH-min's hardness, even in the unconstrained setting, is an open question. Theorem 4.2 resolves this question for the heterogeneous case, showing that it is NP-hard and that no algorithm can do better than a $4/3$-approximation guarantee. The question of hardness in the homogeneous case remains open.

**Theorem 4.2.** *The unconstrained and heterogeneous version of SH-min is NP-hard. Moreover, no poly-time algorithm can achieve an approximation factor better than $4/3$.*

*Proof.* We first show that for any graph $G = (V, E)$ it is possible to construct $f_i$ and $B_i$ such that the corresponding sum in the SH-min problem has minimum value if and only if $A$ is a vertex cover for $G$. For constants $k > 1$ and $\epsilon > 0$, let $\gamma_1 = \frac{2^k - 1}{2^k [(2^k - 1)^{1/k} - 1]^k} + \epsilon$ and $\gamma_2 = \frac{\gamma_1}{2^k - 1}$. For every edge $e = (u, v) \in E$, define two positive polymatroid functions:

$$f_{1e}(Y) = (\gamma_1 |Y \cap u| + \gamma_2 |Y \cap v|)^{1/k}, \quad \text{and} \quad f_{2e}(Y) = (\gamma_2 |Y \cap u| + \gamma_1 |Y \cap v|)^{1/k}. \tag{10}$$

Let $B_{1e} = \{u\}$ and $B_{2e} = \{v\}$ and define the sum $F^c(A)$:

$$F^c(A) = \sum_{e \in E} h_e(A), \text{ for } h_e(A) = f_{1e}(A \triangle B_{1e}) + f_{2e}(A \triangle B_{2e}). \tag{11}$$

The value of each term in this sum is shown in Table 3. Note that the definition of $\gamma_2$ ensures that $(\gamma_1 + \gamma_2)^{1/k} = 2\gamma_2^{1/k}$.

Table 3: Values of $f_{e1}$, $f_{e2}$, and their sum, $h_e$.

| Case | $f_{1e}(A \triangle B_{1e})$ | $f_{2e}(A \triangle B_{2e})$ | $h_e(A)$ |
|---|---|---|---|
| $u \in A, v \notin A$ | $0$ | $(\gamma_1 + \gamma_2)^{1/k}$ | $2\gamma_2^{1/k}$ |
| $u \notin A, v \notin A$ | $\gamma_1^{1/k}$ | $\gamma_1^{1/k}$ | $2\gamma_1^{1/k}$ |
| $u \in A, v \in A$ | $\gamma_2^{1/k}$ | $\gamma_2^{1/k}$ | $2\gamma_2^{1/k}$ |
| $u \notin A, v \in A$ | $(\gamma_1 + \gamma_2)^{1/k}$ | $0$ | $2\gamma_2^{1/k}$ |

Using Table 3, we can show that the minimizers of $F^c(A)$ are exactly the set covers of $G$:

- Case 1—show that every vertex cover of $G$ is a minimizer of $F^c$: By the definition of $\gamma_2$, we know $\gamma_1 > \gamma_2$, and so the minimum value of $F^c$ occurs when all $h_e$ are $2\gamma_2^{1/k}$, which is clearly achievable by setting $A = V$. Any set $A$ that is a vertex cover contains at least one endpoint of each edge, and hence also has value $2\gamma_2^{1/k}$ for each $h_e$.

- Case 2—show that every minimizer of $F^c$ is a vertex cover of $G$: Suppose that $A^*$ is a minimizer of $F^c$ but not a vertex cover of $G$. Then there exists some uncovered edge $e = (u, v)$ with neither endpoint in $A^*$. Consider adding $u$ to $A^*$ to form a set $A'$. The corresponding difference in $h_e$ value is: $h_e(A^*) - h_e(A') = 2\gamma_1^{1/k} - 2\gamma_2^{1/k} > 0$. The difference in $h$-value for each other edge $e'$ that touches $u$ is similarly $2\gamma_1^{1/k} - 2\gamma_2^{1/k}$ if $e'$ is uncovered in $A^*$, or 0 if $e'$ is covered by $A^*$. All other $h$-values remain unchanged. Thus, $F^c(A') < F^c(A^*)$, contradicting the assumption that $A^*$ is a minimizer of $F^c$.

Borrowing from [14]'s Theorem 3.1, we now define a particular graph and two additional positive polymatroid functions. Consider the bipartite graph $G = (V_1 \cup V_2, E)$ where $|V_1| = |V_2| = r$ and the edge set consists of $r$ edges that form a perfect matching of $A$ to $B$. Let $R$ be a random minimum-cardinality vertex cover of $G$. Define the following two functions:

$$f_0^a(Y) = \min\{|Y|, r\} \qquad f_0^b(Y) = \min\left\{|Y \cap \bar{R}| + \min\left\{|Y \cap R|, \frac{(1+\delta)r}{2}\right\}, r\right\} \qquad (12)$$

where $\delta$ is set so that $2/(1 + \delta) = 2 - \epsilon$. [14] shows that, knowing $G$ but given only value-oracle access to the $f_0$, no poly-time algorithm can distinguish between $f_0^a$ and $f_0^b$. Moreover, if restricted to vertex cover solutions, it is easy to see that the function $f_0^a$ is minimized on any of the $2^r$ possible vertex covers, for which it has value $r$, while the function $f_0^b$ is minimized on the set $Y = R$, for which it has value $\frac{(1+\delta)r}{2}$. The ratio of these minimizers is $2 - \epsilon$, which allows [14] to show that no poly-time algorithm can achieve a $(2 - \epsilon)$-approximation for the minimum submodular vertex cover problem.

Now, instead of explicitly restricting to vertex cover solutions, consider unconstrained minimization on $F^a(A) = f_0^a(A) + F^c(A)$ and $F^b(A) = f_0^b(A) + F^c(A)$. Since $f_0^a$ and $f_0^b$ cannot be distinguished in poly-time, neither can $F^a$ and $F^b$. We can also show that: (1) any minimizer of $F^a(A)$ or $F^b(A)$ must be a vertex cover, and (2) the ratio of the corresponding vertex cover minimizers is $4/3$.

- Show $F^a$'s minimizers are vertex covers: Suppose that $A^*$ is a minimizer of $F^a$ but not a vertex cover of $G$. Then there exists some uncovered edge $e = (u, v)$ with neither endpoint in $A^*$. Consider adding $u$ to $A^*$ to form a set $A'$. As shown above, the corresponding difference in $F^c$ value is $2\gamma_1^{1/k} - 2\gamma_2^{1/k}$. The difference $f_0^a(A^*) - f_0^a(A')$ is $-1$ if $|A^*| < r$ and 0 otherwise. Thus, all we need is for $2\gamma_1^{1/k} - 2\gamma_2^{1/k}$ to be $> 1$. Plugging in the definition of $\gamma_1$ and $\gamma_2$, this inequality can be seen to hold for all $k > 1$. Thus, overall $F^a(A') < F^a(A^*)$, contradicting the assumption that $A^*$ is a minimizer of $F^a$.

- Show $F^b$'s minimizers are vertex covers: The reasoning here is analogous to the $F^a$ case; the difference $f_0^b(A^*) - f_0^b(A')$ is always $> -1$, since adding a single node can never change the $f_0^b$ value by more than 1.

- $F^a$'s minimum value: Any vertex cover $A$ includes at least $r$ nodes and thus has value $f_0^a(A) = r$. Since there are $r$ edges total, $F^c(A) = 2r\gamma_2^{1/k}$ for a vertex cover. Combining these we see that $F^a$ has minimum value $r(1 + 2\gamma_2^{1/k})$.

- $F^b$'s minimum value: The vertex cover consisting of the set $R$ minimizes $f_0^b$: $f_0^b(R) = \frac{(1+\delta)r}{2}$. Thus, the minimum $F^b$ value is $r\left(\frac{(1+\delta)}{2} + 2\gamma_2^{1/k}\right)$.

Letting $k \to \infty$, we have that $\gamma_2^{1/k} \to 1/2$. Thus, in the limit the as $k \to \infty$, the ratio of minimizers is: $2/(\frac{(1+\delta)}{2} + 1) = \frac{4}{3+\delta}$. Plugging in the definition of $\delta$ from above, the ratio in terms of $\epsilon$ is: $\frac{4-2\epsilon}{3-\epsilon} > \frac{4}{3} - \frac{2\epsilon}{3-\epsilon} = \frac{4}{3} - o(1)$. $\qquad \square$

Since unconstrained SH-min is NP-hard, it makes sense to consider approximation algorithms for this problem. We first provide a simple 2-approximation, UNION-SPLIT (see Algorithm 1). This algorithm splits $f(A \triangle B) = f((A \setminus B) \cup (B \setminus A))$ into $f(A \setminus B) + f(B \setminus A)$, then applies standard submodular minimization (see e.g. [15]) to the split function. Theorem 4.3 shows that this algorithm is a 2-approximation for SH-min. It relies on Lemma 4.2.1, which we state first.

**Lemma 4.2.1.** *Let $f$ be a positive monotone subadditive function. Then, for any $A, B \subseteq V$:*

$$f(A \triangle B) \le f(A \setminus B) + f(B \setminus A) \le 2f(A \triangle B). \tag{13}$$

*Proof.* The upper bound follows from the definition of $\triangle$ and the fact that $f$ is subadditive:

$$f(A \triangle B) = f((A \setminus B) \cup (B \setminus A)) \le f(A \setminus B) + f(B \setminus A). \tag{14}$$

The lower bound on $2f(A \triangle B)$ follows due to the monotonicity of $f$: $f(A \setminus B) \le f(A \triangle B)$ and $f(B \setminus A) \le f(A \triangle B)$. Summing these two inequalities gives the bound. $\qquad\square$

**Theorem 4.3.** UNION-SPLIT *is a 2-approximation for unconstrained SH-min.*

*Proof.* An SH-min instance seeks the minimizer of $F(A) = \sum_{i=1}^{m} f_i(A \triangle B_i)$. Define $\bar{F}(A) = \sum_{i=1}^{m} [f_i(A \setminus B_i) + f_i(B_i \setminus A)]$. From Lemma 4.2.1, we see that $\min_A \bar{F}(A)$ is a 2-approximation for $\min_A F(A)$ (any submodular function is also subadditive). Thus, if $\bar{F}$ can be minimized exactly, the result is a 2-approximation for SH-min. Exact minimization of $\bar{F}$ is possible because $\bar{F}$ is submodular in $A$. The submodularity of $\bar{F}$ follows from the fact that submodular functions are closed under restriction, complementation, and addition (see [16], page 9). These closure properties imply that, for each $i$, $f_i(A \setminus B_i)$ and $f_i(B_i \setminus A)$ are both submodular in $A$, as is their sum. $\qquad\square$

Note that UNION-SPLIT's 2-approximation bound is tight; there exists a problem instance where exactly a factor of 2 is achieved. More concretely, consider $V = \{1, 2\}$, $B_1 = \{1\}$, $B_2 = \{2\}$, and $f_1(Y) = f_2(Y) = |Y|^{(1/\alpha)}$ for $\alpha > 1$. Then according to the $F'$ passed to SUBMODULAR-OPT, all solutions have value 2. Yet, under the true $F$ the solutions $\{1\}$ and $\{2\}$ have the better (smaller) value $2^{(1/\alpha)}$. Letting $\alpha \to \infty$, the quantity $2^{(1/\alpha)}$ approaches 1, making the ratio between the correct solution and the one given by UNION-SPLIT possibly as large as 2.

Restricting to the homogeneous setting, we can provide a different algorithm that has a better approximation guarantee than UNION-SPLIT. This algorithm simply checks the value of $F(A) = \sum_{i=1}^{m} f(A \triangle B_i)$ for each $B_i$ and returns the minimizing $B_i$. We call this algorithm BEST-B (Algorithm 2). Theorem 4.4 gives the approximation guarantee for BEST-B. This result is known [17], as the proof of the guarantee only makes use of metricity and homogeneity (not submodularity), and these properties are common to much other work. We provide the proof in our notation for completeness though.

**Theorem 4.4.** *For $m = 1$,* BEST-B *exactly solves unconstrained SH-min. For $m > 1$,* BEST-B *is a $\left(2 - \frac{2}{m}\right)$-approximation for unconstrained homogeneous SH-min.*

*Proof.* Define $F(A) = \sum_{i=1}^{m} f_i(A \triangle B_i)$, for $f_i$ positive polymatroid. Since each $f_i$ is normalized and positive, each is minimized by $\emptyset$: $f_i(\emptyset) = 0$. Thus, any given $f_i(A \triangle B_i)$ is minimized by setting $A = B_i$. For $m = 1$, this implies that SH-min is exactly solved by setting $A = B_1$.

Now consider $m > 1$ and the homogeneous setting where there is a single $f$: $f_i = f \;\forall i$. By Theorem 3.1, $f(A \triangle B_i)$ is a metric, so it obeys the triangle inequality:

$$f(A \triangle B_i) + f(A \triangle B_j) \ge f(B_i \triangle B_j) \quad \forall i, j. \tag{15}$$

Fixing some $i$ and summing this inequality over all $j \ne i$:

$$\sum_{j \ne i} [f(A \triangle B_i) + f(A \triangle B_j)] \ge \sum_{j \ne i} f(B_i \triangle B_j) = \sum_{i=1}^{m} f(B_i \triangle B_j) \tag{16}$$

where the last equality is due to the fact that polymatroids are normalized: $f(B_i \triangle B_i) = f(\emptyset) = 0$. Regrouping terms, $f(A \triangle B_i)$ is independent of $j$, so it can be pulled out of the summation:

$$(m - 2)f(A \triangle B_i) + \sum_{j=1}^{m} f(A \triangle B_j) \ge \sum_{j=1}^{m} f(B_i \triangle B_j). \tag{17}$$

**Algorithm 1** UNION-SPLIT

    **Input**: $\mathcal{F}, \mathcal{B}$
    Define $f_i'(Y) = f_i(Y \setminus B_i) + f_i(B_i \setminus Y)$
    Define $F'(Y) = \sum_{i=1}^{m} f_i'(Y)$
    **Output**: SUBMODULAR-OPT $(F')$

**Algorithm 2** BEST-B

    **Input**: $F, \mathcal{B}$
    $A \leftarrow B_1$
    **for** $i = 2, \ldots, m$ **do**
      **if** $F(B_i) < F(A)$: $A \leftarrow B_i$
    **Output**: $A$

**Algorithm 3** MAJOR-MIN

    **Input**: $\mathcal{F}, \mathcal{B}, \mathcal{C}$
    $A \leftarrow \emptyset$
    **repeat**
      $c \leftarrow F(A)$
      Set $\boldsymbol{w}_{\hat{F}}$ as in Equation 23
      $A \leftarrow$ MODULAR-MIN $(\boldsymbol{w}_{\hat{F}}, \mathcal{C})$
    **until** $F(A) = c$
    **Output**: $A$

Notice that $\sum_{j=1}^{m} f(A \triangle B_j)$ is exactly $F(A)$ and $\sum_{j=1}^{m} f(B_i \triangle B_j)$ is $F(B_i)$. Substituting in this notation and summing over all $i$:

$$\sum_{i=1}^{m} [(m-2)f(A \triangle B_i) + F(A)] \geq \sum_{i=1}^{m} F(B_i). \tag{18}$$

On the left-hand side we can again replace the sum with $F(A)$, yielding: $2(m-1)F(A) \geq \sum_{i=1}^{m} F(B_i)$. Since a sum over $m$ items is larger than $m$ times the minimum term in the sum, the remaining sum here can be replaced by a min:

$$2(m-1)F(A) \geq m \min_{i \in \{1, \ldots, m\}} F(B_i). \tag{19}$$

The left-hand size is exactly what the BEST-B algorithm computes, and hence the minimizing $B_i$ found by BEST-B is a $(2 - 2/m)$-approximation for unconstrained homogeneous SH-min. $\qquad\square$

Note that as a corollary of this result, in the case when $m = 2$, the optimal solution for unconstrained homogeneous SH-min is to take the best of $B_1$ and $B_2$. Also note that since UNION-SPLIT's 2-approximation bound is tight, BEST-B is theoretically better in terms of worst-case performance in the unconstrained setting. However, UNION-SPLIT's performance on practical problems is often better than the BEST-B's, as many practical problems do not hit upon this worst case. For example, consider the case where $V = \{1, 2, 3\}$, $f$ is simply cardinality, $f(A) = |A|$, and each $B_i$ consists of two items: $B_1 = \{1, 2\}, B_2 = \{1, 3\}, B_3 = \{2, 3\}$. Then the best $B_i$ has $F$-value 4, while the set $\{1, 2, 3\}$ found by UNION-SPLIT has a lower (better) $F$-value of 3.

### 4.2 Constrained setting

In the constrained setting, the SH-min problem becomes more difficult. Essentially, all of the hardness results established in existing work on constrained submodular minimization applies to the constrained SH-min problem as well. Theorem 4.5 shows that, even for a simple cardinality constraint and identical $f_i$ (homogeneous setting), not only is SH-min NP-hard, but also it is hard to approximate with a factor better than $\Omega(\sqrt{n})$.

**Theorem 4.5.** *Homogeneous SH-min is NP-hard under cardinality constraints. Moreover, no algorithm can achieve an approximation factor better than $\Omega\left(\frac{\sqrt{n}}{1+(\sqrt{n}-1)(1-\kappa_f)}\right)$, where $\kappa_f = 1 - \min_{j \in V} \frac{f(j|V \setminus j)}{f(j)}$ denotes the curvature of $f$. This holds even when $m = 1$.*

*Proof.* Let $m = 1$ and $B_1 = \emptyset$. Then under cardinality constraints, SH-min becomes $\min_{A:|A| \geq k} f(A)$. Corollary 5.1 of [18] establishes that this problem is NP-hard and has a hardness of $\Omega(\frac{\sqrt{n}}{1+(\sqrt{n}-1)(1-\kappa_f)})$. $\qquad\square$

We can also show similar hardness results for several other combinatorial constraints including matroid constraints, shortest paths, spanning trees, cuts, etc. [18, 14]. Note that the hardness established

in Theorem 4.5 depends on a quantity $\kappa_f$, which is also called the *curvature* of a submodular function [19, 18]. Intuitively, this factor measures how close a submodular function is to a modular function. The result suggests that the closer the function is being modular, the easier it is to optimize. This makes sense, since with a modular function, SH-min can be exactly minimized under several combinatorial constraints. To see this for the cardinality-constrained case, first note that for modular $f_i$, the corresponding $F$-function is also modular. Lemma 4.5.1 formalizes this.

**Lemma 4.5.1.** *If the $f_i$ in SH-min are modular, then $F(A) = \sum_{i=1}^{m} f_i(A\triangle B_i)$ is also modular.*

*Proof.* Any normalized modular function $f_i$ can be represented as a vector $\boldsymbol{w}_i \in \mathbb{R}^n$, such that $f_i(Y) = \sum_{j\in Y} w_i(j) = \boldsymbol{w}_i^{\top} \mathbf{1}_Y$. With $Y = A\triangle B_i$, this can be written:

$$f_i(A\triangle B_i) = \boldsymbol{w}_i^{\top} \left[ \mathbf{1}_{B_i} + \mathrm{diag}(\mathbf{1}_{V\setminus B_i} - \mathbf{1}_{B_i})\mathbf{1}_A \right] \tag{20}$$

$$= \sum_{j\in B_i} w_i(j) + \sum_{j\in A}(-1)^{\mathbb{1}(j\in B_i)} w_i(j). \tag{21}$$

Summing over $i$ and letting $C = \sum_{i=1}^{m} \sum_{j\in B_i} w_i(j)$ represent the part that is constant with respect to $A$, we have:

$$F(A) = C + \sum_{j\in A}(-1)^{\mathbb{1}(j\in B_i)} w_i(j). \tag{22}$$

Thus, $F$ can be represented by offset $C$ and vector $\boldsymbol{w}_F \in \mathbb{R}^n$ such that $F(A) = C + \sum_{j\in A} w_F(j)$, with entries $w_F(j) = \sum_{i=1}^{m}(-1)^{\mathbb{1}(j\in B_i)} w_i(j)$. This is sufficient to prove modularity. (For optimization purposes, note that $C$ can be dropped without affecting the solution to SH-min.) $\square$

Given Lemma 4.5.1, from the definition of modularity we know that there exists some constant $C$ and vector $\boldsymbol{w}_F \in \mathbb{R}^n$, such that $F(A) = C + \sum_{j\in A} w_F(j)$. From this representation it is clear that $F$ can be minimized subject to the constraint $|A| \geq k$ by choosing as the set $A$ the items corresponding to the $k$ smallest entries in $w_F$. Thus, for modular $f_i$, or $f_i$ with small curvature $\kappa_{f_i}$, such constrained minimization is relatively easy.

Having established the hardness of constrained SH-min, we now turn to considering approximation algorithms for this problem. Unfortunately, the UNION-SPLIT algorithm from the previous section requires an efficient algorithm for submodular function minimization, and no such algorithm exists in the constrained setting; submodular minimization is NP-hard even under simple cardinality constraints [20] (although see [21] that shows it is possible to get solutions for a subset of the cardinality constraints). Similarly, the BEST-B algorithm breaks down in the constrained setting; its guarantees carry over only if all the $B_i$ are within the constraint set $\mathcal{C}$. Thus, for the constrained SH-min problem we instead propose a majorization-minimization algorithm. Theorem 4.6 shows that this algorithm has an $O(n)$ approximation guarantee, and Algorithm 3 formally defines the algorithm.

Essentially, MAJOR-MIN proceeds by iterating the following two steps: constructing $\hat{F}$, a modular upper bound for $F$ at the current solution $A$, then minimizing $\hat{F}$ to get a new $A$. $\hat{F}$ consists of superdifferentials [22, 23] of $F$'s component submodular functions. We use the superdifferentials defined as "grow" and "shrink" in [24]. Defining sets $S, T$ as $S = V \setminus j, T = A\triangle B_i$ for "grow", and $S = (A\triangle B_i) \setminus j, T = \emptyset$ for "shrink", the $\boldsymbol{w}_{\hat{F}}$ vector that represents the modular $\hat{F}$ can be written:

$$w_{\hat{F}}(j) = \sum_{i=1}^{m} \begin{cases} f_i(j \mid S) \text{ if } j \in A\triangle B_i \\ f_i(j \mid T) \text{ otherwise,} \end{cases} \tag{23}$$

where $f(Y \mid X) = f(Y \cup X) - f(X)$ is the gain in $f$-value when adding $Y$ to $X$. We now state the main theorem characterizing algorithm MAJOR-MIN's performance on SH-min.

**Theorem 4.6.** MAJOR-MIN *is guaranteed to improve the objective value, $F(A) = \sum_{i=1}^{m} f_i(A\triangle B_i)$, at every iteration. Moreover, for any constraint over which a modular function can be exactly optimized, it has a $\left( \max_i \frac{|A^*\triangle B_i|}{1+(|A^*\triangle B_i|-1)(1-\kappa_{f_i}(A^*\triangle B_i))} \right)$ approximation guarantee, where $A^*$ is the optimal solution of SH-min.*

*Proof.* We first define the full "grow" and "shrink" superdifferentials:

$$m_{A,1}^f(Y) \triangleq f(A) - \sum_{j \in A \setminus Y} f(j \mid V \setminus j) + \sum_{j \in Y \setminus A} f(j \mid A), \quad \text{and} \tag{24}$$

$$m_{A,2}^f(Y) \triangleq f(A) - \sum_{j \in A \setminus Y} f(j \mid A \setminus j) + \sum_{j \in Y \setminus A} f(j \mid \emptyset). \tag{25}$$

When referring to either of these modular functions, we use $m_A^f$. Note that the $m_A^f$ upper-bound $f$ in the following sense: $m_A^f(Y) \geq f(Y) \; \forall Y \subseteq V$, and $m_A^f(A) = f(A)$.

MAJOR-MIN proceeds as follows. Starting from $A^0 = \emptyset$ and applying either "grow" or "shrink" to construct a modular approximation to $F$ at $\emptyset$ yields the following simple surrogate function for each $f_i$: $\hat{f}_i(Y) = \sum_{j \in Y} f_i(j)$. The below bound then holds (from [18]):

$$f_i(Y) \leq \hat{f}_i(Y) \leq \frac{|Y|}{1 + (|Y| - 1)(1 - \kappa_{f_i}(Y))} f_i(Y), \; \forall Y \subseteq V. \tag{26}$$

Let $\hat{A} = \operatorname{argmin}_{A \in \mathcal{C}} \sum_{i=1}^m \hat{f}_i(A \triangle B_i)$. Also, let $A^* = \operatorname{argmin}_{A \in \mathcal{C}} \sum_{i=1}^m f_i(A \triangle B_i)$. Then, it holds that:

$$\sum_{i=1}^m f_i(\hat{A} \triangle B_i) \leq \sum_{i=1}^m \hat{f}_i(\hat{A} \triangle B_i) \tag{27}$$

$$\leq \sum_{i=1}^m \hat{f}_i(A^* \triangle B_i) \tag{28}$$

$$\leq \frac{|A^*|}{1 + (|A^*| - 1)(1 - \kappa_{f_i}(A^*))} \sum_{i=1}^m f_i(A^* \triangle B_i) \tag{29}$$

The first inequality follows from the definition of the modular upper bound, the second inequality follows from the fact that $\hat{A}$ is the minimizer of the modular optimization, and the third inequality follows from Equation 26. We now show that MAJOR-MIN improves the objective value at every iteration:

$$\sum_{i=1}^m f_i(A^{t+1} \triangle B_i) \leq \sum_{i=1}^m m_{A^t \triangle B_i}^{f_i}(A^{t+1} \triangle B_i) \leq \sum_{i=1}^m m_{A^t \triangle B_i}^{f_i}(A^t \triangle B_i) = \sum_{i=1}^m f_i(A^t \triangle B_i). \tag{30}$$

$\square$

While MAJOR-MIN does not have a constant-factor guarantee (which is possible only in the unconstrained setting), the bounds are not too far from the hardness of the constrained setting. For example, in the cardinality case, the guarantee of MAJOR-MIN is $\frac{n}{1+(n-1)(1-\kappa_f)}$, while the hardness shown in Theorem 4.5 is $\Omega\left(\frac{\sqrt{n}}{1+(n-1)(1-\kappa_f)}\right)$.

# 5 Maximization of the submodular Hamming metric

We next characterize the hardness of SH-max (the diversification problem) and describe approximation algorithms for it. We first show that all versions of SH-max, even the unconstrained homogeneous one, are NP-hard. Note that this is a non-trivial result. Maximization of a monotone function such as a polymatroid is not NP-hard; the maximizer is always the full set $V$. But, for SH-max, despite the fact that the $f_i$ are monotone with respect to their argument $A \triangle B_i$, they are not monotone with respect to $A$ itself. This makes SH-max significantly harder. After establishing that SH-max is NP-hard, we show that no poly-time algorithm can obtain an approximation factor better $3/4$ in the unconstrained setting, and a factor of $(1 - 1/e)$ in the constrained setting. Finally, we provide a simple approximation algorithm which achieves a factor of $1/4$ for all settings.

**Theorem 5.1.** *All versions of SH-max (constrained or unconstrained, heterogeneous or homogeneous) are NP-hard. Moreover, no poly-time algorithm can obtain a factor better than $3/4$ for the unconstrained versions, or better than $1 - 1/e$ for the cardinality-constrained versions.*

*Proof.* We first show that homogeneous unconstrained SH-max is NP-hard. We proceed by constructing an $F$ that can represent any symmetric positive normalized (non-monotone) submodular function. Maximization is NP-hard for this type of function, since it subsumes the MAX-CUT problem. Hence, the reduction to unconstrained SH-max suffices to show NP-hardness.

Consider an instance of SH-max with $m = 2$ and $B_1 = \emptyset, B_2 = V$:

$$\max_{A \subseteq V} F(A) = \max_{A \subseteq V} f(A) + f(V \setminus A). \tag{31}$$

Given a symmetric positive normalized submodular function $h$, define:

$$f(A) = h(A) - \sum_{i \in A} h(i \mid V \setminus i), \tag{32}$$

where $h(i \mid V \setminus i)$ is short for $h(V) - h(v \setminus i)$. To see that $f$ is a positive polymatroid function, first recall that a symmetric set function is one for which $h(A) = h(V \setminus A) \; \forall A \subseteq V$. Thus, $h(i \mid V \setminus i) = h(V) - h(V \setminus i) = h(\emptyset) - h(i) = -h(i)$. This implies that $f(A) = h(A) + \sum_{i \in A} h(i)$, which is clearly a positive polymatroid. Now, notice that:

$$F(A) = f(A) + f(V \setminus A) = h(A) + h(V \setminus A) + \sum_{i \in V} h(i) = 2h(A) + \sum_{i \in V} h(i). \tag{33}$$

Hence, given an instance of symmetric submodular function maximization, we can transform it into an instance of SH-max, with $F$ defined as above; since $\sum_{i \in V} h(i)$ is a constant, it does not affect which set is the maximizer. Thus, unconstrained SH-max is NP-hard.

To show the hardness of approximation, we borrow a proof technique from Theorem 4.5 of [25]. The idea is to construct two symmetric submodular functions, $h_1$ and $h_2$, which are *indistinguishable*. That is, any randomized algorithm would require an exponential number of calls to the value oracles to tell $h_1$ and $h_2$ apart. The construction of [25] suggests that for both these functions, $h_1(i) = h_2(i) = n - 1$, and hence the constant $\sum_{i \in V} h(i) = n(n - 1)$. Thus, we can write $F_1(A) = 2h_1(A) + n(n-1)$ and $F_2(A) = 2h_2(A) + n(n-1)$. Since $h_1$ and $h_2$ are indistinguishable, so are $F_1$ and $F_2$. Moreover, according to [25], the maximum value of $h_1(A)$ is $n^2/4$, while that of $h_2(A)$ is $n^2/2$. Hence, $F_1$'s maximum value is $n^2/2 + n^2 - n = 3n^2/2 - n$, and $F_2$'s is $n^2 + n^2 - n = 2n^2 - n$. The ratio of these is: $(3n/2 - 1)/(2n - 1) = 3/4 + o(1)$. Thus, no poly-time algorithm can achieve a factor better than $3/4$.

Finally, we establish the hardness of approximation for a cardinality-constrained version of SH-max. In this case, let $m = 1$ and $B_1 = \emptyset$. This SH-max instance is exactly the problem of monotone submodular maximization subject to cardinality constraint, which is not only NP-hard but has a hardness of $1 - 1/e$ [26]. $\square$

We turn now to approximation algorithms. For the unconstrained setting, Lemma 5.1.1 shows that simply choosing a random subset, $A \subseteq V$ provides a $1/8$-approximation in expectation.

**Lemma 5.1.1.** *A random subset is a $1/8$-approximation for SH-max in the unconstrained (homogeneous or heterogeneous) setting.*

*Proof.* This result follows from the fact that a random subset is a $1/4$-approximation for the problem of unconstrained non-monotone submodular maximization [25, Theorem 2.1], and that the non-monotone submodular function $\bar{F}(A) = \sum_{i=1}^{m} [f_i(A \setminus B_i) + f_i(B_i \setminus A)]$ is within a factor 2 of the $F(A)$ of SH-max (see Lemma 4.2.1). Thus, a random set is a $1/4$-approximation for $\max_A \bar{F}(A)$ and a $1/8$-approximation for $\max_A F(A)$. $\square$

An improved approximation guarantee of $1/4$ can be shown for a variant of UNION-SPLIT (Algorithm 1), if the call to SUBMODULAR-OPT is a call to a SUBMODULAR-MAX algorithm. Theorem 5.2 makes this precise for both the unconstrained case and a cardinality-constrained case. It might also be of interest to consider more complex constraints, such as matroid independence and base constraints, but we leave the investigation of such settings to future work.

**Theorem 5.2.** *Maximizing $\bar{F}(A) = \sum_{i=1}^{m} (f_i(A \setminus B_i) + f_i(B_i \setminus A))$ with a bi-directional greedy algorithm [27, Algorithm 2] is a linear-time $1/4$-approximation for maximizing $F(A) = \sum_{i=1}^{m} f_i(A \triangle B_i)$, in the unconstrained setting. Under the cardinality constraint $|A| \leq k$, using the randomized greedy algorithm [28, Algorithm 1] provides a $\frac{1}{2e}$-approximation.*

Table 4: mV-ROUGE averaged over the 14 datasets (± standard deviation).

| HM | SP | TP |
|---|---|---|
| $0.38 \pm 0.14$ | $0.43 \pm 0.20$ | $\mathbf{0.50 \pm 0.26}$ |

Table 5: # of wins (out of 14 datasets).

| HM | SP | TP |
|---|---|---|
| 3 | 1 | **10** |

*Proof.* $\bar{F}(A)$ is a (non-monotone) submodular function that is within a factor 2 of $F(A)$ (see Lemma 4.2.1). The bi-directional greedy algorithm [27, Algorithm 2] provides a $1/2$-approximation to non-monotone submodular maximization in the unconstrained setting. Thus, applying it to $\bar{F}$ yields a $1/4$-approximation for $\max_A F(A)$. Similarly, in the cardinality-constrained setting, one can use the randomized greedy algorithm [28, Algorithm 1], which has a $1/e$ approximation guarantee. □