[Reviews · NeurIPS 2015]

Submitted by Assigned_Reviewer_1

026

"the key" -> "one key"

-- there might be more Sec2 Maybe give an example for f for structure prediction and diverse m-best as done for clustering.
Summary: In the paper a subclass of discrete metrics is specified which are fairly tractable.

Submitted by Assigned_Reviewer_2

First, I must say that the paper was very well written and was a joy to read.

The authors analyse metrics of the form d(A, B) = F(A \triangle B), where F is a positive polymatroid function. As the authors state, it was known that even a weaker condition is sufficient for metricity, and the authors analyse the benefits of the additional submodular assumption. The resulting analysis is for both minimizing and maximizing the sum of distances. The algorithms mostly rely on approximating F(A \trinagle B) by F(A \ B) + F(B \ A), which is submodular in A and is a 2-approximation (as they prove). Moreover, this seems to hold even when f is only subadditive and positive, not necessarily submodular. Can the authors please clarify this?

The theoretical results seem correct and they are clearly presented. I went only through some of the proofs, and they were also well written and easily understandable.

The authors propose two new algorithms for minimization. It is a bit disappointing that in the homogeneous case, they could not improve on the existing algorithm B-Bound, which makes a much weaker assumption. I would have been much happier if there were experimental results for minimization as well, as it is not clear if this is due to the looseness of the bound of Union-Split. The proposed algorithm (Major-Min) for the constrained case is very natural as the same strategy (minimize the upper bound from a supergradient) has been already used.

For maximization, the authors suggest the same approximation strategy and then their results follow from the 2-approximation. I believe that it would be more instructive if the authors present the fact that the maximization of any positive symmetric submodular function is a special case for m=2 in the main paper.

Here are some questions I have regarding the experimental section:

* You claim that the greedy algorithm has a 1 - 1/e approximation guarante for HM,

but adding the |A \triangle A_i| term destroys the monotonicity of the objective.

Isn't this a mistake?

* Did you try experimenting with alpha * hamming? I would be interested in the

results under such a simple modification of the baseline model.

* Do you present variance, or standard error in the results? Almost all of the

intervals seem to be significantly overlapping.
Summary: The authors propose a new set of diversity metrics that use submodularity and analyze the resulting optimization problems (note that this is a stronger condition that what is necessary). The theoretical results are sound and well-presented, and the experiments could be improved (for example there is nothing on minimization). I believe that there are enough novel contributions to interest many people at NIPS.

Submitted by Assigned_Reviewer_3

This paper proposed a new Hamming distance metric such that submodular properties can be found from its resulting polymatroid functions. Then, the existing submodular maximization/minimization techniques can be applied here to optimize the Hamming metric.

I have two major questions:

1. All the technical results are based on the assumption that f and g are positive polymatroid functions. I would like to see more discussions about what kind of functions will satisfy the positive polymatroid constraint with examples. Also, for a general problem which doesn't satisfy such a constraint, would it be useful to adopt the metric to pursue an approximate solution?

2. For the diverse summary experiments, the diversity quality is very difficult to evaluate, which makes the results less convincing. Is there any way to better interpret the results? I'd better do the experiments on CRF diverse inference, which can better quantify the results (e.g., top-10 configuration accuracy).

After reading the rebuttal, I am basically satisfied with the clarification and interpretation made by the authors.
Summary: This paper proposed a new Hamming distance metric for a class of polymatroid functions which satisfy submodular properties. The main achievement is tackling the hardness for maximizing and minimizing this Hamming metric. The k-best summary experiments are also given to show one application of the metric. The hardness results are solid, but the experiments are somewhat weak.

Submitted by Assigned_Reviewer_4

The standard Hamming distance (of two sets) is a modular function on their symmetric difference. The paper considers generalisations, where the distance is a monotone submodular function of the symmetric difference. A thorough analysis is given for corresponding optimisation tasks arising e.g. in clustering or for empirical risk minimisation based learning of structured output SVMs. It is important to notice that even the unconstrained versions of the corresponding optimisation tasks are usually hard because a submodular function of the symmetric difference of two sets is not submodular if seen as a function of one of its arguments. The authors analyse the hardness of different task versions and discuss approximation algorithms including their optimality bounds.

The paper is well structured and well written. It seems to be technically correct (I have not checked all proofs given in the supplementary material). The results are partially novel and highly relevant for machine learning. The main contribution is in my view the careful and thorough analysis of the task variants, their hardness and discussion of possible approximation algorithms. I expect that the presented methods will be especially significant for structured output SVM learning, where so far only simple Hamming distances have been used as loss functions.

The presented experiment is in my view somewhat lagging behind the theoretical potential of the paper. Seeking 15 summaries, each consisting of 10 images, from a collection of 100 images is too simple for demonstrating the significance of the proposed methods. I would have expected a convincing experiment for structured output SVM learning instead.
Summary: The paper analyses submodular extensions of Hamming metrics and their applicability for clustering, learning of structured output SVMs and diverse k best solutions of submodular optimisation tasks. A careful analysis of tractability and approximation algorithms is given for the corresponding constrained minimisation and maximisation tasks.

Author Feedback
Author rebuttal: We thank the reviewers for their thoughtful suggestions and the time taken to read our paper. The main concern raised seems to be lack of experiments to complement the theory presented in the paper. In the submitted draft of the paper, our experiments focus on the maximization case, specifically the application of diverse summarization. However, following the deadline, we also have some new experiments to show the performance of the algorithms for the minimization variant (SH-min). In particular, we explore the clustering application described in Section 2. We plan to add these results if the paper gets accepted.

Below, we address concerns raised by individual reviewers.

Reviewer_1:

1) Indeed, the decomposition \bar{F}(A) = \sum_i f(A/B_i) + f(B_i/A) is a two-approximation to the true objective F(A) = \sum_i f(A \triangle B_i) whenever F is positive and subadditive (not necessarily submodular). However the function \hat{F}(A) is easy to minimize or maximize *only* if F is submodular. Hence the approximation guarantee of the Union-Split *only* holds if the function is submodular. (The paragraph following Theorem 3.1 (lines 193-200) in the paper gives more details about the hardness of optimizing subadditive functions.) We will further clarify this in the paper.

2) The Union-Split algorithm's 2-approximation bound is tight; there exists a problem instance where exactly a factor of 2 is achieved. (We will include this example in the paper to clarify.) This implies that the Best-B algorithm is theoretically better in terms of worst-case performance in the unconstrained setting. However, it is also true that Union-Split's performance on practical problems can be better than the Best-B's, as many practical problems do not hit upon this worst case. For example, consider the case where V = {1,2,3}, f is simply cardinality, f(A) = |A|, and each B_i consists of two items: B_1 = {1,2}, B_2 = {1,3}, B_3 = {2,3}. Then the best B_i has F-value 4, while the set {1,2,3} found by Union-Split has a lower (better) F-value of 3.

3) Table 1 gives a bound of 1 - 1/e as the *hardness* of SH-max. Note that the approximation guarantees are in Table 2, and are weaker. This hardness result is established in Theorem 5.1 (proof included in the supplementary material).

4) We could certainly try adding a hyperparameter alpha to experiment with alpha * hamming as an additional summarization method. We will try to incorporate this suggestion in the final version of the paper.

5) The numbers presented after the +/- in Table 4 are standard deviations. While it is true that there is significant overlap between the numbers in Table 4, the results in Table 5 provide some additional support for the superiority of the submodular method.

Reviewer_2:

1) A number of natural choices of diversity and coverage functions are polymatroidal. Examples of these include set cover, facility location, saturated coverage, and concave-over-modular functions. This is a very broad class of submodular functions, which come up in many applications. We will try to describe these in more detail in the paper introduction. It would be interesting future work to extend our algorithms to an even larger class of functions, for instance the non-monotone submodular class. However, approximation guarantees for this larger class do not follow immediately, at least for the algorithms presented in this work. Extending to such a class will likely require more complex algorithms to yield comparable approximation guarantees.

2) In addition to the qualitative example in Figure 1, as a quantitative indicator of the goodness of the summary, we provide the v-ROUGE scores. This is a measure for image collection summarization developed and tested in prior work (see Tschiatschek et al, NIPS 2014).

Reviewer_3: As future work, we would indeed like to evaluate our algorithms for learning structured prediction (this is in fact, one of our motivating applications from Section 2).

Reviewer_5: Thank you for the suggestion; we will extend Section 2 to provide details of choices of submodular functions that one could use in the structured prediction and diverse k-best applications.